

# #UbirajaraBelongstoBR: social media activism

# against (neo)colonial practices in palaeontology

Mohammad Ali Rahimi Fard Kashani[1*], Nussaïbah B. Raja[2], Chico Q. Camargo[134]

[1]Department of Computer Science, University of Exeter, Exeter, EX4 4QJ, United Kingdom
[2]GeoZentrum Nordbayern, Department of Geography and Geosciences, Friedrich-Alexander University Erlangen-Nürnberg, Erlangen, Germany
[3]Department of English Language and Literature, Ewha Womans University, Seoul, South Korea
[4]Alan Turing Institute, London, United Kingdom

*Correspondence to*: M.A Rahimi Fard Kashani (rahimi2709@gmail.com)





# #UbirajaraBelongstoBR: social media activism against (neo)colonial practices in palaeontology

**Abstract.**

Social media has revolutionized the engagement between scientists and the public, offering platforms to challenge unethical practices and advocate for change. In December 2020, Brazilian paleontologists and supporters initiated the hashtag #UbirajaraBelongstoBR on Twitter (now X) to protest the alleged illicit acquisition and export of the dinosaur fossil *Ubirajara jubatus* from Brazil to Germany. This movement not only demanded the fossil's repatriation but also sparked global discussions on neo-colonial practices in paleontology.

In this study, we analyze 39,728 tweets containing the hashtag #UbirajaraBelongstoBR, collected between December 2020 and February 2023. Employing social network analysis and computational text analysis, we examine the dynamics of this online movement, identify key influencers, and assess its reach and impact. Our results reveal that the campaign transcended the paleontology community, engaging a diverse international audience including scientists, artists, activists, and the general public. Sentiment analysis indicates shifts corresponding to pivotal events, such as official statements and the eventual repatriation of the fossil.

Our findings demonstrate the power of social media in mobilizing grassroots movements and influencing scientific discourse and policy. The #UbirajaraBelongstoBR case illustrates how digital platforms can facilitate international advocacy against unethical scientific practices, highlighting social media's potential to effect change in scientific governance and promote ethical standards. This study contributes to the understanding of digital activism in science communication and underscores the evolving landscape of public engagement in scientific issues.

**Keywords** — Social Network Analysis, Ubirajara, Palaeontology, Social Movement, illicit fossil traffic.



# 1 Introduction

Social media has transformed the way in which scientists communicate their research and communication with both other scientists and non-scientists (Ocon et al., 2021; Entradas et al., 2020; Walter et al., 2019). Social media is also serving an important role in providing a platform, especially for marginalised voices in academia, who are harnessing its power to challenge existing power structures (Yammine et al., 2018). It has also been used to call out unethical practices and other issues in the academy. For example, Elisabeth Bik who is known for posting on wrongful image manipulation in the biomedical literature, that has led to the retraction of several papers, has amassed more than 130,000 followers on Twitter (now X) (Bik, 2022; Shen, 2020).

In a similar fashion, Brazilian palaeontologists took to social media in December 2020 to raise their concerns about a newly described Brazilian dinosaur *Ubirajara jubatus* (henceforth Ubirajara). Originating from the Araripe Basin, north east of Brazil, Ubirajara has now come to represent the fight against colonial practices in modern palaeontology. The article in which Ubirajara was described raised several ethical and legal concerns over the appropriation and study on this fossil: 1) the authors and the German institution where it was reposited claimed to have obtained the fossil legally although there did not seem to be any legal avenue for such a fossil to have found its way to Germany; 2) the authors did not seem to have involved any Brazilian institutions in the process of acquiring and researching this specimen—also against the law; 3) The region of Brazil where the fossil originated has been targeted by fossil smugglers for years; (4) Some authors in the study had been involved in controversies with irregular fossils from Brazil before (Cisneros et al., 2022a, b; Christakou, 2015; Gibney, 2014; Raja and Dunne, 2023).

The hashtag #UbirajaraBelongstoBR, protesting for the return of the fossil to Brazil, took Twitter by storm, along with other social media platforms, such as Facebook, Instagram and YouTube, bringing together both Brazilian and non-Brazilian palaeontologists, paleoartists, students and other members of the general public together. It also fuelled discussions on colonialism in palaeontology, a topic of growing interest and importance in the discipline, e.g. (Monarrez et al., 2021; Raja et al., 2022). Palaeontology as a discipline has been shaped through centuries of colonial practices, influencing not only the distribution of fossil data around the world, but also the people who have access to these data (Monarrez et al., 2021; Raja et al., 2022). These practices, including the theft of fossils and exclusion of local scientists as in the case of Ubirajara (Cisneros et al., 2022a), remain to this day and continue to deepen this bias (Raja et al., 2022). The modern materialisation of colonialism in



68 palaeontology is especially directed at the lower and middle income countries which are disproportionately
69 underrepresented in paleontological research and literature (Raja et al., 2022).

70

**Figure 1: Examples of images shared with the hashtag #UbirajarabelongstoBR by Twitter users. Image credits (from left to right, top to bottom): @arturvict (Artur, 2020), @_themingau (Mingau, 2020), @Emily_Art (Stepp, 2020), @herbertologist (Herbert, 2020), @valent801 (Valent801, 2020), @Brenda7Kauane (Kauane, 2021), @freakyraptor (Alli, 2020), @Waxosaurus (Waxosaurus, 2020), @THSpike (Paleonecromante, 2020), @antoniopedroalb (Albuquerque, 2021).**

Ubirajara is not the only Brazilian fossil studied by foreign researchers that is believed to have been illegally
exported and/or acquired. Brazil, especially the Araripe Basin, has been the victim of fossil smuggling for
decades, e.g. holotypes of Irritator challengeri described in 1996 and Mirischia asymmetrica in 2004 (Cisneros et
al. 2022a ; Cisneros et al. 2022b). These have however not attracted as much attention from the public and the
media until Ubirajara (e.g.), probably because these were published before the "golden age" of social media and
science communication on these platforms.
In the case of Ubirajara, social media was crucial in the dissemination of information on the specimen and
updates on the case. Eventually, the State Museum of Natural History Karlsruhe (SMNK) took to Instagram





posting a statement that the fossil was the property of the German state of Baden-Wurttemberg and it would not
repatriate it in response to the backlash it was receiving and attracting more criticism. The corresponding research
article was permanently withdrawn soon after this when it was found out that the authors lied and misrepresented
information on how it was acquired. The Minister of Science of Baden-Wurttemberg, after an investigation,
recognized misconduct by SMNK, declared that the export of the fossil to Germany violated Brazilian laws, and
that the museum should repatriate it to Brazil, which has now happened (Black 2022). Since the
#UbirajaraBelongsToBR movement, several other Brazilian fossils, from the US and Belgium, have been
repatriated (Black 2022).

The #UbirajaraBelongsToBR case shares commonalities with other recent social movements with large online
participation, such as Black Lives Matter (BLM), #MeToo, and the Occupy Wall Street movement. Like these
movements, it leverages the reduced coordination costs and enhanced organizational modes provided by social
and digital technologies. Scholars like Earl and Kimport (Earl and Kimport, 2011) have highlighted how these
technologies not only amplify existing forms of activism but also create fundamentally new ways of operating
within social movements. They argue that digital infrastructures personalize online content, giving individuals
specific reasons to protest and facilitating their ability to push for change across various platforms. This
transformation has demanded a new framework of understanding, as it shifts how activists organize,
communicate, and interact in efforts to achieve widespread societal impact.

Considering the similarity between the #UbirajaraBelongsToBR protest and the other ones cited above, it is
natural to study them using similar approaches. For the Occupy Wall Street anti-capitalist movement in the
United States, for instance, the movement on Twitter appeared to draw a group of people who were already
involved in local politics and other social movements abroad and who were well-connected. Conover et al
(2013a; 2013b) use a sizable sample from Twitter to track trends in Occupy member activity, interests, and
socialisation over a period of fifteen months, starting three months before the regime's first resistance
movements. They find that users who were vocal in the early months of the movement decreased their
involvement in Occupy-related activity during the analysis (Conover et al., 2013a). For comparison, related
studies looking at the expression of the Black Lives Matter movement on Twitter found that BLM activity on
Twitter predicted mainstream news coverage of police brutality, which in turn was the strongest driver of
attention to the issue from political elites (Freelon et al. 2016). At the same time, Ince et al. show how that BLM
was not a monolithic movement with a single message or way to frame structural racism issues, but rather a





movement with "distributed framing" (Ince, Rojas, and Davis 2017), where different hashtags co-occurring with #BlackLivesMatter emphasized different aspects of the movement, such as solidarity towards those protesting, strategic tactics, violent reactions from the police, counter-movement sentiment, among others.

Here, taking inspiration from studies such as the ones mentioned above, we analyse a collection of all tweets containing the hashtag #UbirajaraBelongsToBR during a given period, and investigate the how the #UbirajaraBelongsToBR movement evolved over time, in terms of its tone, language, and groups of participants, as important milestones in the case of Ubirajara happened over many months. Finally, our results indicate that the #UbirajaraBelongstoBR managed to burst the palaeontology bubble, and to go even beyond science and science journalism.

## 2 Materials and Methods

### 2.1 Data

Tweets posted during the period December 2020 and February 2023 that contain the hashtag #UbirajaraBelongsToBR were downloaded using the Twitter API (now unavailable, after the rebrand to X) using the python library Tweepy (Roesslein 2009).

The final dataset includes 39,728 tweets that included the hashtag #UbirajaraBelongstoBR and related metadata such as the user account from which the tweet was posted, the tweet type, the data and time, the number of likes and retweets, as well as other information. The preprocessing stage involved the removal of duplicate tweets, along with the conversion of all hashtag text to lowercase. By adopting this approach, we ensured the inclusion of identical hashtags (e.g. #UbirajaraBelongsToBR and #ubirajarabelongstobr). The presence of repeated hashtags within a single tweet.

Ethical clearance for this project was obtained at the University of Exeter, including for the publication of user identifiers (e.g. twitter handles), provided that the data did not include any sensitive content. Since it did not include any non-public or sensitive content, the project was approved.

### 2.2 Language detection

For tweets with undefined languages, the Google Translate API was used to detect the language of the original tweet and or referenced tweet. When that failed, the detected language by Google Translate API has been used as the final value, by priority of Original Tweet language rather than Referenced Tweet language.





## 2.3    Country detection

Over 90% of the returned data by Tweepy listed an undefined country. We used a combination of tools such as pycountry, CountryInfo, and the Google Maps v3 API to infer the country of residence of the users. This was done by using the Location feature which is filled by users in their user accounts. This feature includes different types of data such as addresses, cities, or maybe countries, written in full or as abbreviations (e.g. BRA for Brazil). Countries could only be detected for approximately 50% of the tweets, as many users do not provide any country-specific information in their profile. For the users with undefined countries, we only tagged the language of their tweets. For instance, a user with an undefined country posting primarily in Portuguese was set as "Unknown (Portuguese)".

## 2.4    Sentiment analysis

We also carried out a sentiment analysis for each tweet using Natural Language Processing (NLP) to analyse the polarity in opinions, sentiments and feelings expressed in each tweet. This was done using the function SentimentIntensityAnalyzer() from the Natural Language Toolkit (NLTK) in Python. The foundation of sentiment analysis is a lexicon that associates lexical traits with emotional intensity scores. The intensity of each word in a text may be added together to determine the sentiment score of that text. Words like "loving," "joy," "glad," and "like" carry positive connotations whereas  terms, such as that "did not love" is deemed as a negative statement.

NLTK returns a list of scores  for each of the following four parameters for a string:: Negative, Neutral, Positive and Compound (calculated by averaging the preceding scores). For instance, for 'This was a good movie.' the result will be: ['neg': 0.0, 'neu': 0.508, 'pos': 0.492, 'compound': 0.4404]. In essence,  this tool can detect the intensity of positivity and negativity of each phrase according to their words and punctuation (Kumar et al. 2022). It is worth noting that the SentimentIntensityAnalyzer tool is limited in its ability to detect nuanced forms of conversations such as irony or sarcasm, but assuming that the majority of the tweets in our dataset do not fall in that category, it should still produce useful results.

## 2.5    Network analysis

We use Gephi, an open source tool for manipulating networks (Bastian et al. 2009), to analyse the interaction between users and explore the structure of their connection as well as the connections amongst #UbirajaraBelongsToBR and co-occurring hashtags. This allows us to identify the structure of interactions and





attributes of members and possible patterns within them, to recognise regional and global patterns, significant
people, and network dynamics.
Since we focus on studying interactions between individuals, we only considered tweets that were either a
retweet, a quoted tweet or a reply to the original tweet and built a directed network that considers the direction of
communication, e.g..
User A has replied to User B by a tweet. Each interaction between two specific users was assigned a value
depending on the number of times it happened, e.g. if user A retweets a tweet from user B for 5 times, a weight of
5 was assigned.
We carry out community detection using the Louvain algorithm (Blondel et al., 2008). A community in a network
can be roughly defined as a group of nodes more densely connected to each other than to nodes outside the group.
For a network inferred from posts and hashtags, these communities are centred around a topic of conversation, or
based on shared interests or attributes. Finally, to estimate the centrality of authors in the protest network, we use
multiple network science statistics, defined below.

**Degree centrality**: this is the simplest measure of centrality and counts the number of edges (or connections) a
node has. Nodes with a high degree centrality are often hubs or highly connected nodes in a network. In a social
network, for example, a person with a high degree centrality might have many friends.

**Betweenness Centrality**: this measure looks at all the shortest paths between pairs of nodes and counts how
often a particular node lies on these paths. Nodes with high betweenness centrality act as bridges or gateways in
the network. They are often crucial for ensuring flow or connectivity between different parts of the network.

**Eigenvector Centrality:** this centrality measure assigns relative scores to all nodes in the network based on the
idea that connections to high-scoring nodes contribute more to the score of a node than equal connections to low-
scoring nodes. Nodes with high eigenvector centrality are connected to many nodes who themselves have high
scores. It's a measure of "influential" connectivity.
Each centrality measure gives a different perspective on the importance or influence of nodes within a network.
Depending on the research question or the nature of the network, one might be more relevant than the others.





## 3 Results

### 3.1 Protest timeline



**Figure 2: Timeline of events during the #UbirajaraBelongstoBR protest.**

In press article on "Ubirajara jubatus" appears on the journal Cretaceous Research and the hashtag #UbirajaraBelongstoBR is first used on Twitter.
The Brazilian Society of Palaeontology contacts Cretaceous Research.
First report on the controversy by international media (National Geographic).
Article temporarily removed by Cretaceous Research.
SMNK informs the Brazilian Society of Palaeontology that it will not repatriate the fossil. The Brazilian Society of Palaeontology informs its members.
SMNK releases a statement on Instagram refusing repatriation of "Ubirajara jubatus".
A petition is created at Change.org asking for the fossil to be returned.
Article withdrawn by Cretaceous Research.
SMNK Instagram account is deactivated.
Article in the journal Science reveals that the dinosaur was imported to Germany by fossil dealers in 2006 and purchased by SMNK in 2009.
The USA repatriates 35 fossil spiders to Brazil.
Belgium repatriates a pterosaur to Brazil.
Germany announces that "Ubirajara jubatus" will be repatriated to Brazil.

There are two significant moments (Figure 2) with a high number of tweets over the selected time duration, namely the start of the protest and the first report on the controversy by international media, and the moment



when SMNK informed the Brazilian Society of Palaeontology that it would not repatriate the forum, both of
which led to many manifestations on twitter. Specific dates and links to each event are provided in Appendix A.



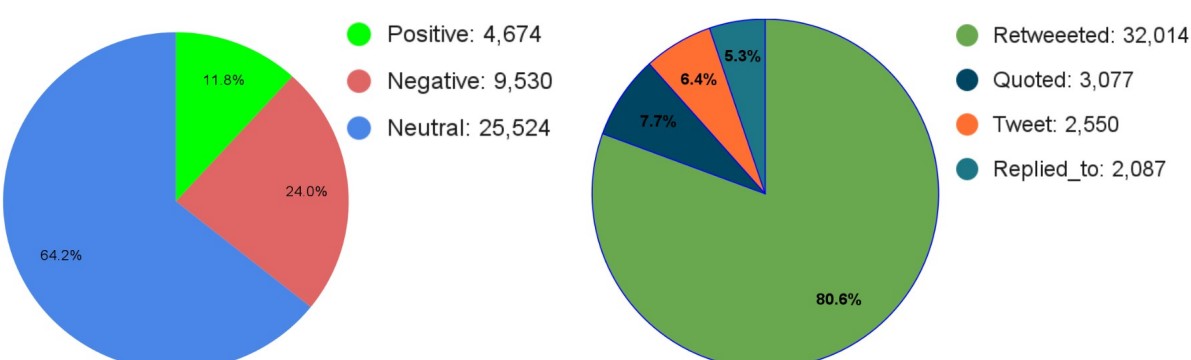

**Figure 3: (left) Distribution of tweet sentiment (right) Distribution of Tweet groups**

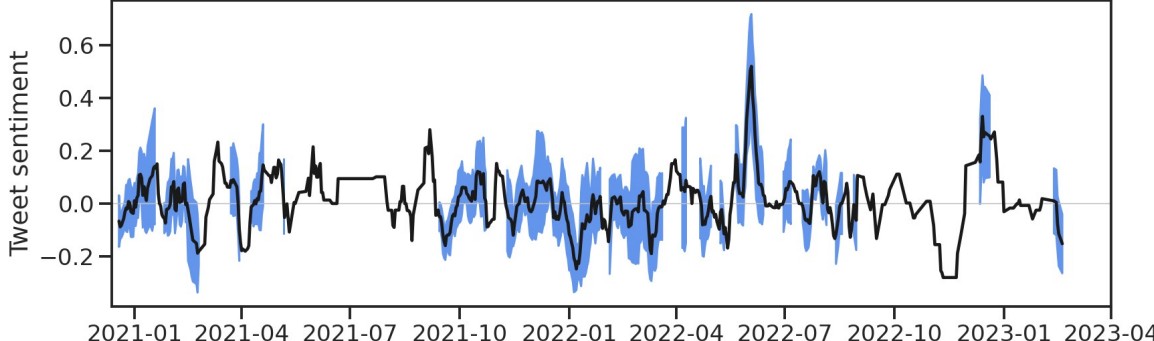


**Figure 4: Overall tweet sentiment over time. The black line represents the average sentiment of all tweets**
**containing #UbirajaraBelongstoBR, and the shaded blue area represents the average plus or minus one**
**standard deviation of the tweet sentiment.**

The sentiment analysis results (Figures 3 and 4) reveal that while a majority of tweets show neutral sentiment, the
number of negative tweets is more than two times the number of positive tweets. They also show how most of the
tweets related to #UbirajaraBelongstoBR were retweets, i.e. amplified material from other accounts. The overall
sentiment is on average neutral, as shown more clearly over time (Figure 4). There are several moments of larger





standard deviation on the number of tweets, marked by peaks of higher positive sentiment in July 2022, when
Germany announced that the fossil would be repatriated to Brazil.

**3.2    Social network structure of the protest**

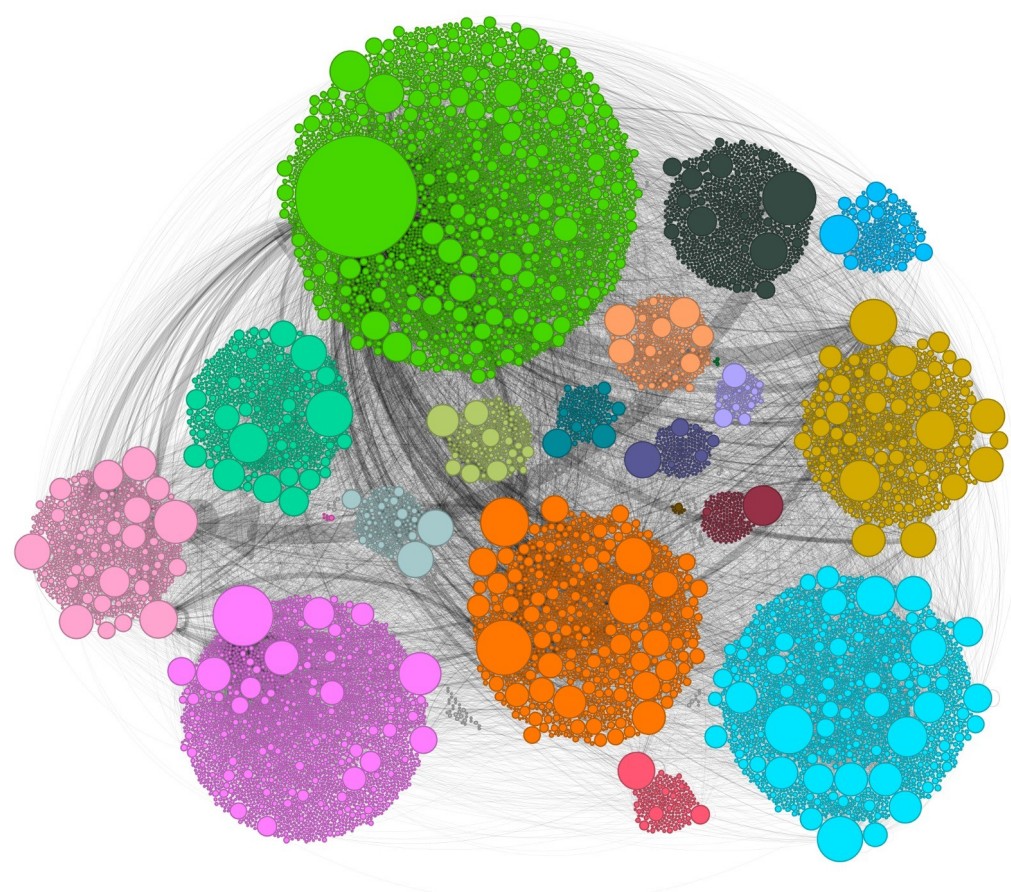


**Figure 5: social network of users participating in #UbirajaraBelongstoBR, connected by their shared**
**interactions. Node colours indicate communities in the network.**

The social network of users participating in #UbirajaraBelongstoBR shows that each community involves one
user as a leader and some others that follow the leader (Figure 5). The size of each node indicates the in-degree of



each user, i.e. the number of people interacting with them, and the colour of each node indicates the community
where each user belongs, as identified by the Louvain community detection algorithm. We find several well-
separated communities of users, which we describe in more detail below.

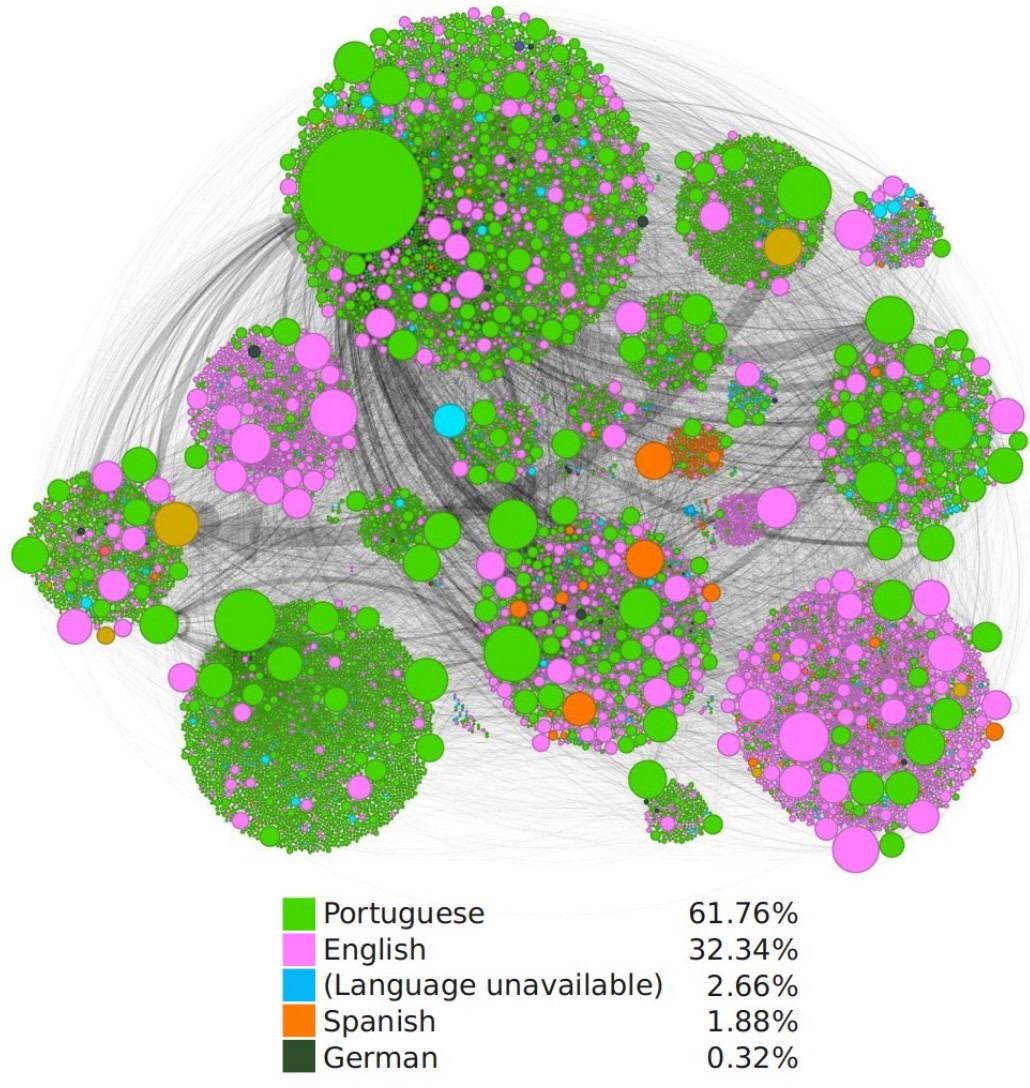

**Figure 6: social network of users participating in #UbirajaraBelongstoBR, connected by their shared interactions. Node colours indicate tweet language.**



Comparing the social networks of users (Figures 5 and 6) reveals that most of the members in each community
share the same language. Most people protesting about Ubirajara were Portuguese speakers as evidenced by
61.76% of the nodes shown in green. There is also a large fraction (32.34%) of users tweeting in English, shown
in pink, and smaller proportions of users tweeting in Spanish, German, and other languages.

| Degree Centrality | Betweenness Centrality | Eigenvector Centrality |
|---|---|---|
| alinemghilardi | alinemghilardi | alinemghilardi |
| BiodiversidadeB | BiodiversidadeB | PaleoCisneros |
| PaleoCisneros | mikannn | MMarcosaurus |
| mikannn | PaleoCisneros | BiodiversidadeB |
| pansybeast | pansybeast | oTroianoleo |

**Table 1: Highest centrality users, as calculated using different centrality metrics.**

We also assess the most central users of the protest network, as measured by the three centrality metrics defined
above, namely degree centrality, betweenness centrality, and eigenvector centrality. Despite the differences
between how each centrality measure is defined, the three measures point at a roughly consistent set of main
actors pushing the conversation: @alinemghilardi, paleontologist and Professor at the Brazilian Federal
University of Rio Grande do Norte, @BiodiversidadeB, a Twitter account covering biodiversity content run by
João Pedro Salgado, @PaleoCisneros, paleontologist and professor at the Brazilian Federal University of Piauí,
@PansyBeast, the account of Julian Francis Miholics, an illustrator who contributed to the protest. The top 5
accounts according to each centrality measure also include @mikannn, corresponding to Miriam Castro,
journalist and pop culture influencer, @MMarcosaurus, corresponding to Marcos K. Pinheiro, geoscience student
and artist, and @oTroianoleo, historian and archaeologist, all of which supported the protest.

The high centrality of paleontologists and biodiversity accounts in the protest network draws into question the
actual reach of the movement. This can be assessed by examining the users present in each community (Figure
7). The figure shows the type of account for the 10 top members of each community which has over 100
members, from community 1 (with 3064 members) to community 16 (with 151 members). Each community
shows a different composition in terms of accounts, as shown by the different colours on the pie charts. While the
largest community in the network is made of mostly palaeontologists, the remaining communities have a diverse



breakdown of other account types, ranging from scientists and science journalists to artists, politicians, institutional accounts, as well as a large fraction of personal user accounts. While it is not possible to examine each and every account present in all the communities on the network, this result suggests that the #UbirajaraBelongstoBR did indeed reach beyond the palaeontology bubble, and even beyond science and science journalism. The full table of the highest degree users per community is shown in Appendix B.



**Figure 7: Type of account for the 10 top members of each community with over 100 members.**



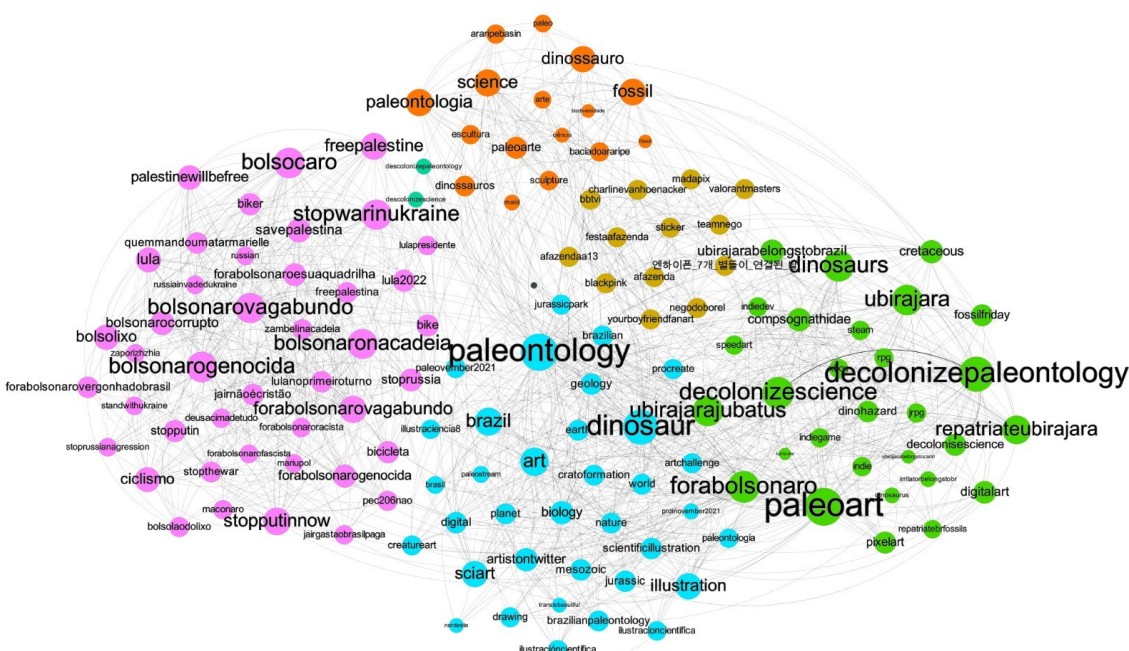


**Figure 8: A figure with the network of co-occurring hashtags, with node colour indicating different**
**communities of hashtags used in the same tweets.**

Finally, hashtags such as #paleontology, #decolonizepaleontology, and #paleoart as central and important nodes
in the main community (Figure 8). Those hashtags, shared by the palaeontologists who led the movement, appear
co-occurring with several minor hashtags, such as #brazil, #art, or #repatriatubirajara. It is also worth noting how
the Ubirajara protest hashtags also appeared alongside other protest hashtags relating to international political
events, such as #stopwarinukraine and #freepalestine, as well as to Brazilian politics, such as
#Bolsonarogenocida, #Bolsonarovagabundo, #Bolsocaro, #Forabolsonaro (meaning "genocidal Bolsonaro",
"Bolsonaro bum", "Bolsonaro expensive", "Bolsonaro out", respectively) and #Lula (Lula da Silva, Bolsonaro's
main opponent at the time and current president of Brazil). This spillover of the protest event towards other
political topics can be interpreted as an indication of the political leaning of the Twitter users who were
concerned about #UbirajaraBelongstoBR.
**4    Discussion and Conclusion**
The #UbirajaraBelongstoBR movement represents a clear example of the intersection of social media activism
and the politics of science and academia, particularly within the realm of paleontology. Our analysis highlights



that individuals with pre-existing social media influence, such as paleontologists and science communicators,
played crucial roles in amplifying the movement. This pre-existing presence provided a platform that effectively
raised awareness and mobilized public opinion, as evidenced in the widespread sharing and engagement with the
hashtag. Notably, the involvement of paleontologists brought authoritative voices into the public discourse,
lending credibility and urgency to the concerns raised about colonial practices in paleontology. At the same time,
the impact of non-paleontologists, including artists and the general public, demonstrates the movement's reach
beyond academic circles, enabling a broader societal engagement.
Our detailed analysis of Twitter data reveals not only the significant role of Portuguese speakers in propelling the
discussion but also underscores the contribution of the English-speaking community in a predominantly
Portuguese discourse. This bilingual dynamic indicates a broader international concern and engagement with
issues of neo-colonial practices in science.
We also find that fluctuations of retweeting and sentiment correlated to important events in the study case. The
first one happens in the period between the release of the *in press* manuscript and its temporary removal by
Cretaceous Research two weeks later (2020-12-24). The hashtag #UbirajarabelongstoBR was widely mentioned
in local media and also appeared on international outlets around this time (Greshko, 2021). The second
fluctuation happens after the announcement by SMNK that the Brazilian dinosaur would not be repatriated
(2021-09-09). This is reflected in the negative sentiment of the tweets. The German museum —who lacks a
Twitter account— published a statement on Instagram and Facebook declaring the Brazilian fossil to be
"property of the State of Baden-Württemberg", generating great backlash on both platforms. Only two weeks
after this announcement, the manuscript was permanently withdrawn by Cretaceous Research and it was revealed
that the authors of the manuscript provided false information regarding the export of Ubirajara (Pérez Ortega,
2021). Finally, the last fluctuation, reflecting positive tweet sentiment, is connected to the announcement (2022-
07-15) by Germany that the dinosaur would be repatriated (Pérez Ortega, 2022).
The Ubirajara case is far from being an isolated one (Cisneros et al., 2022b). The Araripe Basin in Brazil has
been a hub of illegally exported fossils for museums and private collectors since the end of the twentieth century
(Cisneros et al., 2022a). This issue has long been a source of concern for local scientists and authorities, and
echoed by the local press but largely overlooked by both the international scientific community and the foreign
media. An exception to this trend was a report by the journal Nature (Gibney, 2014) on Brazil's efforts to fight
illegal fossil trade, and the case of the snake-like lizard *Tetrapodophis* in 2015 (Christakou, 2015). The latter
represents, to our knowledge, the first instance of an Araripe fossil study whose legal and ethical circumstances



were publicly questioned outside Brazil. This case, however, had only a mild presence on social media and did not achieve the repercussions that Ubirajara had.

As mentioned above, social media activism allows underrepresented voices to attract attention and concentrate discussions around topics that normally would be ignored or misrepresented. The large use of #UbirajaraBelongstoBR on Twitter is a clear example of that, in how it quickly spread and gained attention, collecting efforts from a wide public and forcing the issue to be noticed by stakeholders and news vehicles. This was possible by both the large social media penetration in Brazil and its potential for use as a science communication tool. Some studies point that Brazil is now a leader in science communication on social media platforms, with Facebook, Twitter and Youtube being the main venues (Entradas et al., 2020; Velho and Barata, 2020; Velho et al., 2020). Moreover, the integration of artworks through the #Paleoart among others not only enriched the movement's aesthetic but also broadened its appeal and accessibility, allowing for a more diverse demographic engagement. This inclusion of artistic expressions underscores the multidimensional impact of social media movements, bridging science, art, and activism.

Historical contexts such as the one of the Araripe Basin and the continuous illegal fossil trade emphasize the systemic issues within paleontological research and highlight the need for stricter regulations and more ethical conduct within the scientific community. The significant media coverage that followed the spread of the hashtag #UbirajaraBelongstoBR illustrates the power of social media in bringing international attention to local issues, which have been previously overlooked by global audiences and the scientific community alike. The interconnection of the #UbirajaraBelongstoBR movement with broader political movements against Brazil's far-right government at the time also further contextualizes the social and political dimensions of the protest. This alignment suggests that the movement was not only about scientific and ethical issues but also reflected broader societal and political dissent.

As with any study relying on data collected from social media, our analysis has its limitations: although all tweets with #UbirajaraBelongstoBR from the study period were collected, this study does not include related tweets which did not include this specific hashtag. Tweet language inference is also not 100% accurate, and the anonymity enabled by platforms such as Twitter/X means that even upon close examination it is not always possible to infer a user's language, location, or field of activity (e.g. if they are palaeontologists or not). Finally, after the recent changes in its data access policy, Twitter/X is no longer an easily accessible data source for scientific research on social movements – a phenomenon which reflects the current state of research on online platforms (Freelon et al. 2018)





Still, this study opens potentially fruitful avenues for future research. One potential direction is the examination
of the impact of social media on the governance of scientific research and the enforcement of ethical standards,
whether for scientific research or not. Another area could involve exploring the role of digital activism in shaping
public policy and international agreements on cultural and scientific heritage, such as the policies around illicit
fossil trafficking in the case of *Ubirajara*. Additionally, further studies could investigate the long-term impacts of
such movements on public trust in science and on the practices within the paleontological community.
In conclusion, the #UbirajaraBelongstoBR movement not only challenged neo-colonial practices in paleontology
but also showcased the transformative potential of social media as a tool for global awareness and advocacy. This
case study serves as a testament to the power of collective action through digital platforms and highlights the
evolving landscape of public engagement in scientific discourse.



## 5    Appendix A: Full timetable of events

| Date | Event |
|------|-------|
| 2020-12-13 | *In press* article on "Ubirajara jubatus" appears on the journal Cretaceous Research. [1] |
| 2020-12-13 | The hashtag #UbirajaraBelongstoBR is first used on Twitter by Aline M. Ghilardi. [2] |
| 2020-12-14 | The Brazilian Society of Palaeontology contacts Cretaceous Research. |
| 2020-12-22 | First report on the controversy by international media (National Geographic). [3] |
| 2020-12-24 | Article temporarily removed by Cretaceous Research. |
| 2021-09-08 | SMNK informs the Brazilian Society of Palaeontology that it will not repatriate the fossil. The Brazilian Society of Palaeontology informs its members. |
| 2021-09-09 | SMNK releases a statement on Instagram refusing repatriation of "Ubirajara jubatus". |
| 2021-09-10 | A petition is created at Change.org asking for the fossil to be returned. [4] |
| 2021-09-22? | Article withdrawn by Cretaceous Research. |
| 2021-09-28 | SMNK Instagram account is deactivated. |
| 2021-09-29 | Article in the journal Science reveals that the dinosaur was imported to Germany by fossil dealers in 2006 and purchased by SMNK in 2009. [5] |
| 2021-10-15 | The USA repatriates 35 fossil spiders to Brazil. [6] |
| 2022-02-08 | Belgium repatriates a pterosaur to Brazil. [7] |
| 2022-07-19 | Germany announces that "Ubirajara jubatus" will be repatriated to Brazil. [8] |



**Table A1: Timeline of the #UbirajaraBelongstoBR protest.**



[1] https://www.sciencedirect.com/science/article/pii/S0195667120303736
[2] https://twitter.com/alinemghilardi/status/1338199196348919816?s=20&t=Uv9P5IZAXaSwBFXR1GyYWA
[3] https://www.nationalgeographic.co.uk/science-and-technology/2021/01/one-of-a-kind-dinosaur-removed-
from-brazil-sparks-backlash
[4] https://www.change.org/p/ubirajara-belongs-to-brazil
[5] https://www.science.org/content/article/maned-dinosaur-fossil-will-head-back-to-brazil-after-controversy-
over-import-to-germany
[6] https://www.opovo.com.br/noticias/cariri/2021/10/15/traficada-aranha-fossil-que-homenageia-pablo-vittar-
retorna-ao-cariri.html
[7] https://g1.globo.com/ce/ceara/noticia/2022/02/02/fossil-de-cranio-de-pterossauro-originario-da-bacia-do-
araripe-no-ceara-e-devolvido-ao-brasil-por-museu-da-belgica.ghtml
[8] https://mwk.baden-wuerttemberg.de/de/service/presse/pressemitteilung/pid/land-gibt-dinosaurier-fossil-aus-
naturkundemuseum-karlsruhe-an-brasilien-zurueck-1



## 6 Appendix B: Users with highest degree in each community


| Community | Top 10 members by in-degree |
|---|---|
| 0 | alinemghilardi, PlantaSim, willibrunow, MaximusSpino, schrarstzhaupt, JoanaOrfao, TewBlack, SerpInFormes, kimim01, pedrowisq |
| 1 | boringsuchus, paleoeddye, [suspended user], catalina_leite, pilgrimcetus_, luizacaires3, MatheusKnothe, BRodriguesOhana, o_weverton, MarinesWitzke |
| 2 | WryCritic, MF_gadelha, PPaleoartist, DiAmador4, JuliotheArtist, sadtheropod, [suspended user], _PaleoGeek_, PalaeoVsRacism, LionsDenArtwork |
| 3 | ProjetoCiencia, tito_aureliano, dpaulocarvalho, kalebmelkor, ruzzibarbara, Pirulla25, bioriderjr, RabelloAnderson, eosauria, paleopirata |
| 4 | PaleoCisneros, Machado_DSc, mauritiantales, mathchaos, palaeodaniel, rpocisv, paleoTsimoes, PStewens, Yara_Haridy, RenanBantim |
| 5 | Colecionadores2, _themingau, Joseane_sf, Albertossauro, nishi_kazue, tainancia, PedroHTunes, ArqueoPreHist, CoelhoPre, nigthstrange |
| 6 | FlavianaJorge, JornalOGlobo, xicosa, elis_sntn, [suspended user], VenomaniaKou, revistapiaui, mwk__bw, wolverinegeo, NatGeo |
| 7 | smcarvalho42, giordaness, GabrielBritozz, portsmouthuni, [suspended user], sr_kenway, ikessauro, poeirinhadoalem, capetaman, pauloal97618063 |
| 8 | mikannn, PrazerCambraia, Sybylla_, rogan dopraga, CamaradaHidalgo, _ohcrab, lentevermelha, pifalcao, DiegoCrux, analesnovski |
| 9 | oTroianoleo, dwnews, R0dr1got3, paleorocha, Camila_18FJ, vleonelss, mponcci, |





| | |
|---|---|
| | LutzLeo, AmbBrasilia, dw_brasil |
| 10 | antoniopedroalb, lucaskias, PaleoBlogBR, MarcosTeo2, marciolcastro, pteroana, saradrawspaleo, Hypnos_art, Vinsevla1, THSpike |
| 11 | FeliPinheir, jutyrannus, tylerstoneart, tupaguerra, JersonTatu, DimetroDude, o_eco, almeidacm3, victor_debrito, allen_pancake |
| 12 | DaltonPinheiro1, nenel_leonam, MeioDeCultura, casavoguebrasil, MarcusRibeiroM2, saturns005, Brenin_m_b, rosecoloredjoca, C4iman15, jbubadue |
| 13 | MMarcosaurus, ValeriaRoman, luc14nobio, sgufmg, rraf_aelbio, beccarivictor, Rafa_paleo, ramonsiilvaas, iMalvikaGaur, TomHoltzPaleo |
| 14 | brunobittar91, viadescendens, fadelandia, HaruJiggly, ratgroundpear, Nido_Quing, Lillyywho, badwitchmaris, subjetividdfeia, jinkitopia |
| 15 | BiodiversidadeB, PerboniRenato, MarjorieMbeller, Akamezinha, galileufanacc, LUNAtichenr, paulamariane27, rozzz_zz, folha_ciencia, PetraDeQuartzo |
| 16 | InsetoLand, isisrnd, Leo_Tusi, ttluao, ClelsonFraga, brunojose_, Luigi0131, LyraSid, _themonie_, Bugseelf |
| 18 | PesquisaFapesp, AllBrPolitics, Fenix_glacialis, 2XVIINI, LuanMoldanMotta, hummyeonbird, anai_pari_, Rabiaandrea, robsongfreire, anabee |
| 19 | KerberLeonardo, PsychoAna_xD, PaleoCameron, Jorllyrey, doralcoelho, [suspended user], MosaFabim, sasimarie, edvardvallek, saurianboy |
| 20 | fedkukso, adagamante, HenriqueRandom, peregrino0788, [suspended user], stephanevw, balsedie, centaurus_crux, barroso2501, gustavoburin |


**Table B1: Top 10 members by in-degree for each community in the protest network.**



## 7 Code and data availability

All code and data are available at https://github.com/evoluchico/ubirajarabelongstobr.

## 8 Author contributions

CQC coordinated the project and designed the study. MARFK collected the data and carried out the initial analyses, CQC carried the later ones. MARFK and CQC produced all figures. NBR contributed with the literature review and motivation. MARFK wrote the initial material, CQC produced the first draft of the paper, and all authors gave final approval for publication.

## 9 Competing interests

The authors declare that they have no conflict of interest.

## 10 Ethical statement

The authors confirm the research received ethical clearance from the University of Exeter's Computer Science department.

## 11 Acknowledgements

The authors would like to thank Aline Ghilardi and Juan Cisneros for multiple conversations and clarifications about #UbirajaraBelongstoBR. Additionally, the authors would like to thank Felipe L. Pinheiro for his support labelling data. Finally, the authors would like to thank everyone who participated in the #UbirajaraBelongstoBR, by drawing, posting, sharing, retweeting, and supporting the movement.





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
