# Peer review of "UbirajaraBelongstoBR: social media activism against (neo)colonial practices in palaeontology"

_EGUsphere, 2024_

## Referee Comment (RC1)

[referee-annotated manuscript omitted]

---

## Author Response (AR1)

**Paper:** #UbirajaraBelongstoBR: social media activism against (neo)colonial practices in palaeontology

**Authors:**
Mohammad Ali Rahimi Fard Kashani, Nussaïbah B. Raja, Chico Q. Camargo
* * *
Dear Editor,

Thank you for your careful examination of our paper. We have gone through all the comments from the interactive review, making many changes, described below as response to all comments:

**CC1**: 'Comment on egusphere-2024-3826', Thomas Clements, 17 Jan 2025 reply

Great manuscript - I read it with great interest. I think it would be useful to mention in the intro the account that started the hashtag, and comment on the start of the movement. I think the context of the original tweet would be very useful to know. Otherwise, fantastic paper!

**Citation**: https://doi.org/10.5194/egusphere-2024-3826-CC1

- **AC1**: 'Reply on CC1', Chico Camargo, 21 Jan 2025 reply

Thank you! That's a very good point actually. We'll incorporate that into the text. Thanks for the suggestion!

**Citation**: https://doi.org/10.5194/egusphere-2024-3826-AC1

**RC1**: 'Comment on egusphere-2024-3826', Anonymous Referee #1, 08 Feb 2025 reply

- **AC2**: 'Reply on RC1', Chico Camargo, 13 Feb 2025 reply

Thank you for your comments! We really appreciate all the attention to our text. We'll reply in order:

1. It should be noted that the name 'Ubirajara' is no longer available as a taxonomic name after ZooBank removed it in November 2022 (https://www.mapress.com/zt/article/view/zootaxa.5254.3.10 ). As such, it should be written not in italics to differentiate it from the other valid names in the paper.

    **Reply:** Thank you for this important technical point. We will incorporate this into our draft on the first mention of Ubirajara.

2. This paper also adds to the argument that fossils have a cultural value and are also cultural heritage (reinforced by the art put in Fig. 1 and the many communities it reached). In this sense, I would add that "Ubirajara" had a cultural significance. The name "lord of the spear" is from an ancient Tupi language. I remember some tweets or articles also pointing out that Ubirajara was the main protagonist of a 19th-century novel called "Ubirajara" by Jose de Alencar, whose protagonist was an indigenous warrior not corrupted by European culture. The analysis in this paper frames that thinking of fossils also as cultural heritage could help the community navigate its protection, as argued in the de Araújo-Júnior et al. (2024) (https://www.nature.com/articles/s41559-024-02397-6 ).

   **R:** This is a great point, which even connects well with the fact that artists helped push the hashtag forward. We will add this to the Discussion.

3. It would also be important to highlight that this paper is also an effort to document and archive a social movement that is now obscured after many of the accounts left Twitter following the blocking of Twitter in Brazil in 2024 to encourage more of these types of analyses at a time where there is a conscientious effort to reshape the way scientific societies talk and engage.

   **R:** Indeed we often ignore the importance of documenting and archiving social movements. Thanks for the suggestion. We will incorporate this into the Discussion.

4. I do not think we need stricter rules (line 334). Rather, the current commitments of journals and scientific societies should be enforced. Chacon-Baca et al. (2023) highlighted that even though most journals adhere to the Committee on Publication Ethics (COPE), most do not require specifications on legal or ethical requirements in their submission guides for authors.

   **R:** That's a good point. We find that that's often the case: the rules on

paper are fair and strict enough, but they are not enforced. We've added your point to the discussion, on that same line.

5. It would be helpful to contextualise the ebbs and flows in sentiment over time, including the first or very first tweets, and analyse their sentiment score.

   **R:** We agree with the reviewer, and have now done this analysis. Our results indicate that although the overall average sentiment is neutral over the whole timeline of the protest, points in time when meaningful events happened (which caused more tweets) are also moments with a higher standard deviation on the overall sentiment – typically, lots more positive tweets supporting the movement and celebrating small victories, with equal measure of tweets expressing support not with positivity, but with anger and criticism towards the expropriation of the fossil, or towards colonialism in palaeontology. We will add a detailed analysis of sentiment over time to our text, looking into which tweets show which sentiment, and crossing that with tweet count over time.

6. Finally, I wonder if it is possible to zoom in on the behaviour of the communities found in the network analysis and break down the sentiment analysis. Some interesting questions I have are: Is Community 1 (the top accounts being palaeontologists and scientists) mostly retweeting the top members and, therefore, amplifying the original sentiment of the tweets? Is Community 12 (also mostly scientific but more personal and sci-comm) more of a discussion that combines negative and positive responses?

   Since most of the campaign is retweeting, this would help contextualise the sentiment fluctuation. Also, it would help explain why some communities feel the discussion was mostly negative and others feel it was mostly positive.

   **R:** That's a great suggestion – we have now done this analysis and will add it to the paper. In a nutshell, we find that almost all communities are mostly retweeting, but communities 1, 7, 11, and 12 have mostly original tweets. Community 1 does indeed include the main paleontologists spearheading the movement, but the ratio of original tweets to retweets/quote-tweets is approximately 50%/50%. while communities 7 and 11 mostly had users tagging other users, like "Hey @you look at this! #UbirajaraBelongsToBR". In community 12, surprisingly, over 70% of tweets come from a single user, @bioriderjr, who has since deleted their account. In their tweets, the user was messaging Brazilians politicians, science communicators, the German Ambassador and German Embassy in Brazil, as well as Theresia Bauer, who

at the time headed the Ministry of Science, Research and Arts for the German state of Baden-Württemberg, demanding the repatriation of the fossil. Finally, we find that the breaking sentiment per community does not yield any meaningful results – no community significantly expressed more positive or negative sentiment than any other.

Thank you for inviting me to review this paper. I congratulate the authors on their ideas and efforts to preserve and archive this important milestone in palaeontology, which the disintegration, disappearance or fragmentation of Twitter communities have eroded.

**R:** Thank you so much for your support and critical suggestions! Our paper now has even more interesting results after your input.

**Citation**: https://doi.org/10.5194/egusphere-2024-3826-AC2

**RC2**: 'Comment on egusphere-2024-3826', Anonymous Referee #2, 05 Mar 2025 reply

- **AC3**: 'Reply on RC2', Chico Camargo, 12 Mar 2025 reply

https://egusphere.copernicus.org/preprints/2025/egusphere-2024-3826/#discussion

We thank the reviewer for their comments! Here are the answers to your questions:

1. While reading the results from Figure 6, I became interested in knowing if German-speaking Twitter users were supporting the hashtag. With that it would be interesting for me to know the intersected distribution of the twitter users, for example, those in English are the paleontologists, and those are the most connected nodes ou encountered. In case this implies a very long section of the paper and additional work, I would recommend the authors display the characteristics of the users in Table 1, including language, type of category, and number of followers.

R: Thank you for your points. Regarding the German-speaking users, we have now included this text:

"Looking at German in particular, further analysis reveals a total of 170 tweets in German, all of them supporting the hashtag, with 51 tweets adopting a negative tone, 114 of them a neutral tone, and only 5 tweets adopting a positive tone. We also note that 22 tweets are from users self-reporting a location in Brazil, 14 tweets from users self-reporting a location in Germany, and most tweets either in other countries or not reporting any location."

Regarding the intersection between language and user types, we don't have account type for all users, but we do have that for the top users -- so will edit Table 1 to include language, type of user, and number of followers.

2. In Figure 7. when you show the results by type of account, it should be clarified how the Twitter users were labeled into these categories, and this information should be written in a subsection near the explanation of the language detection.

R: Thank you for raising this methodological point. We have now added text describing that on the part where this figure is described. It now reads as:

"All of these accounts were examined by the authors, and manually categorised into ten categories of users, based on the information available in each account: Palaeontologists, Scientists of other disciplines, Science Journalists, other Science Communicators, Press (e.g. newspapers), other Institutions (e.g. museums), Artists, Politicians, Personal accounts, and Other, which includes deleted users, suspended accounts, and Twitter bots. Each community shows a different composition in terms of accounts, as shown by the different colours on the pie charts."

3. In the discussion, I had some conflicts when reading the following statement: "Our analysis highlights that individuals with pre-existing social media influence, such as paleontologists and science communicators, played crucial roles in amplifying the movement." The reason is that I don't see how the results support this statement. If the data is available, I would suggest creating a table/plot comparing, for example, the degree of the authors in the network with the number of followers; it would be even more accurate if these values could be retrieved at the very beginning of your dataset timeline. I think that information would help to clarify if the most central users were central before this movement and then support your statement or if this movement made them influential.

R: We agree with the reviewer. We now have added information on the degree of the most influential users in the movement, and will revise this statement accordingly:

"Our analysis highlights the crucial role of individuals with pre-existing social media influence in amplifying this movement. Such is the case of the paleontologists, scientists, and science communicators shown in Table 1."

4. Finally, I think the statement of the paper about the power of online activism could be reinforced if there is information provided about public offline manifestations of the Ubirajara case. The online sphere is helping to connect discourses worldwide, but I would like to read in the paper if this is also supported by on-site activism or there was not much and it is driven purely online. On the same line, I would suggest the authors incorporate the work of this book "#HashtagActivism: Networks of Race and Gender Justice" as the discourse in both cases seems aligned.

R: This manifestation was predominantly online. We now discuss that in the

Conclusion. We appreciate the mention of the #HashtagActivism book as well -- and agree it is very aligned with our research. We now cite it as well.

**Citation**: https://doi.org/10.5194/egusphere-2024-3826-AC3

Finally, we also address the editor's comment:

**Editor:** I have not personally worked with social media data. However working in a social sciences capacity, I am aware that individual identifiers should not be included. Therefore I believe any edits to the manuscript or identifiers of individuals (e.g @) should be removed. For example you refer to a specific account in the reply to the reviews.

**Author response:** as a rule of thumb, we agree it is good practice to anonymise user identifiers. However, in this case, as pointed out by the reviewers, this paper is also an effort to document and archive a social movement that is now obscured after many of the accounts left Twitter following the blocking of Twitter in Brazil in 2024. In other words, this is a case where usernames do matter – in particular those who were pivotal in making #UbirajaraBelongsToBR successful. And since we believe the paper is not putting anyone in harm's way, or exposing people in an unnecessary manner, we believe the best choice is to follow the reviewers' suggestions and include relevant usernames. Finally, as for the artists whose artwork is presented in Figure 1, we reached out to the artists for consent to include their usernames.